# Identifying Molecular Pathophysiology and Potential Therapeutic Options in Iatrogenic Tracheal Stenosis

**DOI:** 10.3390/biomedicines12061323

**Published:** 2024-06-14

**Authors:** Russell Seth Martins, Joanna Weber, Bryan Johnson, Jeffrey Luo, Kostantinos Poulikidis, Mohammed Jawad Latif, Syed Shahzad Razi, Al Haitham Al Shetawi, Robert S. Lebovics, Faiz Y. Bhora

**Affiliations:** 1Division of Thoracic Surgery, Department of Surgery, Hackensack Meridian School of Medicine, Hackensack Meridian Health (HMH) Network, Edison, NJ 08820, USA; russellseth.martins@hmhn.org (R.S.M.); joanna.weber@hmhn.org (J.W.); jeffrey.luo@hmhn.org (J.L.); kostantinos.poulikidis@hmhn.org (K.P.); mohammed.latif@hmhn.org (M.J.L.); syed.razi@hmhn.org (S.S.R.); 2Department of Surgery, Mount Carmel Health System, Columbus, OH 43213, USA; bryan.johnson001@mchs.com; 3Division of Surgical Oncology, Department of Surgery, Vassar Brothers Medical Center, Nuvance Health, Dyson Center for Cancer Care, Poughkeepsie, NY 12601, USA; al-haitham.al-shetawi@nuvancehealth.org; 4Division of Oral and Maxillofacial Surgery, Department of Surgery, Vassar Brothers Medical Center, Nuvance Health, Poughkeepsie, NY 12601, USA; 5Division of Otolaryngology, Department of Surgery, Hackensack Meridian School of Medicine, Hackensack Meridian Health (HMH) Network, Edison, NJ 08820, USA; robert.lebovics@hmhn.org

**Keywords:** subglottic stenosis, transcriptomics, next-generation sequencing, gene ontology, retinoids

## Abstract

Introduction: While most patients with iatrogenic tracheal stenosis (ITS) respond to endoscopic ablative procedures, approximately 15% experience a recalcitrant, recurring disease course that is resistant to conventional management. We aimed to explore genetic profiles of patients with recalcitrant ITS to understand underlying pathophysiology and identify novel therapeutic options. Methods: We collected 11 samples of granulation tissue from patients with ITS and performed RNA sequencing. We identified the top 10 most highly up- and down-regulated genes and cellular processes that these genes corresponded to. For the most highly dysregulated genes, we identified potential therapeutic options that favorably regulate their expression. Results: The dysregulations in gene expression corresponded to hyperkeratinization (upregulation of genes involved in keratin production and keratinocyte differentiation) and cellular proliferation (downregulation of cell cycle regulating and pro-apoptotic genes). Genes involved in retinoic acid (RA) metabolism and signaling were dysregulated in a pattern suggesting local cellular RA deficiency. Consequently, RA also emerged as the most promising potential therapeutic option for ITS, as it favorably regulated seven of the ten most highly dysregulated genes. Conclusion: This is the first study to characterize the role of hyperkeratinization and dysregulations in RA metabolism and signaling in the disease pathophysiology. Given the ability of RA to favorably regulate key genes involved in ITS, future studies must explore its efficacy as a potential therapeutic option for patients with recalcitrant ITS.

## 1. Introduction

Tracheal stenosis occurs most commonly as a sequela of iatrogenic causes such as prolonged intubation and tracheostomy [1]. While over 50% of patients experience some degree of tracheal insult after prolonged intubation [2], the incidence of iatrogenic tracheal stenosis (ITS) is reported to range from 6 to 21% [3]. ITS can be a devastating and chronic disease for patients to endure, necessitating long-term management, and can negatively impact both physical and psychosocial quality of life [4].

ITS is extremely challenging to manage, with surgical and endoscopic interventions being the mainstay of treatment. These include endoscopic balloon dilation, laser therapy, spray cryotherapy, tracheal stenting, and, less commonly, tracheal resection and reconstruction. However, surgery is limited by several contraindications, including previous surgery or mucosal inflammation beyond the area of resection, as well as by a patient’s general operability [5]. While endoscopic interventions provide a reasonable alternative to open surgery, relapse rates are higher [6]. Thus, patients often require routine intervention for symptom control. Several adjunctive medical therapies such as glucocorticoids and antibiotics have been explored to consolidate interventional management, but with limited success [7,8].

While the pathogenesis of ITS is not completely understood, literature suggests that underlying genetic susceptibilities and immune dysregulations in the setting of tracheal insult may lead to the unregulated hypergranulation seen in ITS. Molecules such as VEGF (vascular endothelial growth factor), PDGF (platelet-derived growth factor receptor), FGF (fibroblast growth factor), and TGF-β (transforming growth factor-beta) have been identified as contributors to the disease process. Subsequently, these molecules have been therapeutic targets with medications such as pirfenidone, nintedanib, and rapamycin, which have shown some promising results in animal models [9]. Prior work with coronavirus disease 2019 (COVID-19)-associated tracheal stenosis (CATS) further corroborates the potential role for genetic and immune dysregulations in the development of tracheal hypergranulation [10]. In CATS, genes associated with anti-viral and anti-microbial functions was upregulated, such as CXCL11 (C-X-C motif chemokine ligand 11), CCL8 (C-C motif chemokine ligand 8), DEFB103A (defensin beta 103A), IFI6 (interferon alpha inducible protein 6), ACOD1 (aconitate decarboxylase 1), and DEFB4A (defensin beta 4A) [10]. Based on these dysregulations, several drugs with specific gene product targets have been suggested as potential therapies for CATS [10]. However, gene expression analysis of ITS granulation tissue is relatively nascent and an understanding of the molecular mechanisms behind ITS may hold promise in terms of novel treatment and preventive options. Thus, we aimed to investigate patterns of gene expression in the tracheal granulation tissue of patients with ITS and identify potential therapeutic options based on genetic profiles.

## 2. Materials and Methods

This study was conducted between July 2019–July 2021 at Nuvance Health in Connecticut, USA, after receiving ethical approval from the institutional review board (ID: 2019-19).

### 2.1. Patient Enrollment and Sample Collection

Eight adult patients (age ≥ 18 years) presenting to clinics for management of severe, recalcitrant ITS, as visualized by upper airway endoscopy, were included in this study after informed consent was obtained. Inclusion criteria included over 70% tracheal lumen occlusion and at least two recurrences which required cryotherapy/balloon dilation procedures. Patient data collected included demographics, baseline health status, and clinical history.

To understand active disease mechanisms in ITS (prior to potentially irreversible tissue scarring), we examined transcriptomic dysregulation in granulation tissue. Tracheal tissue biopsies of granulation tissue were collected at the time of endoscopic intervention, which consisted of balloon dilation and spray cryotherapy. If enrolled patients returned to clinics requiring reintervention for re-stenosis, they were reapproached for collection of additional samples. Additional biopsies were collected from consenting patients, with these being considered as unique samples for analysis.

### 2.2. RNA Sequencing of Samples

Frozen tissue samples were sent to Azenta Life Sciences for RNA extraction, processing, and RNA sequencing. cDNA libraries were generated using Illumina preparation kits with Poly(A) selection. Sequencing was conducted using the Illumina HiSeq^®^ platform configured to the following settings: 2 × 150 bp, ~350 M paired-end reads, single index, per lane with ≥80% of bases ≥Q30. Analysis was performed using the vendor standard RNA-seq analysis.

### 2.3. Normal Control

Publicly accessible genomic data for normal tracheal tissue was sourced from the public data repositories of the National Center for Biotechnology Information (NCBI: SRR16760102) and the European Nucleotide Archive (ENA: ERR 2022844).

### 2.4. Data Analysis

CLC Genomics Workbench by QIAGEN was used for analysis of the sequenced data. Trimmed reads were aligned and annotated with Ensembl 91: December 2017 (GRCh38.p10). Differential gene expressions were explored between tissue samples of ITS and normal control data. Gene expression was considered significant if the FDR (false discovery rate) *p*-value was <0.5 and the fold change (ratio of value in specimen to value in normal control) was >1.5. The top 10 most highly upregulated and downregulated genes were identified by calculating a change coefficient that accounted for both FDR *p*-value and fold change, as follows: |*change coefficient| = −log*_2_
*FDR p-value × log*_2_
*fold change*. The protein class of the gene products and their relevant functions were retrieved from the Human Protein Atlas [11], an open-source repository containing data on all proteins coded by the human genome. For each of these pathologically dysregulated genes, we queried the Human Portal of the Rat Genome Database [12] to identify potential therapeutic options that could inhibit pathologically upregulated gene products or promote pathologically downregulated gene products. We prioritized drugs with FDA approval for use in humans.

Gene Ontology (GO) and KEGG (Kyoto Encyclopedia of Genes and Genomes) functional enrichment were applied via DAVID (Database for Annotation, Visualization, and Integrated Discovery) for significantly upregulated and downregulated genes separately. Functional annotations, enriched GO terms, and pathways identified were noted. Only pathways and processes with a Benjamini *p*-value <0.05 were considered significant.

## 3. Results

A total of 11 unique samples from the 8 patients were sequenced for analysis. Table 1 shows the ten most upregulated and downregulated genes (by change coefficient) in our ITS tissue samples. Amongst the upregulated genes, the greatest fold changes were seen in *DEFB103A* (fold increase: 10.24), *DSG1* (8.52), and *CASP14* (8.47). Amongst the downregulated genes, the greatest fold decreases were seen in *C16orf82* (fold decrease: 125,487.85), *PRB4* (10,386.22), and *DPEP1* (2730.08). Figure 1 shows a volcano plot depicting the differential expression of all sequenced genes in ITS tissue samples compared to the normal controls. Figure 2 shows a heatmap depicting expression of the most highly upregulated and downregulated genes in each individual tissue sample. The *KRT16*, *KRT14*, and *KRT6B* were most consistently upregulated across samples, while *C16orf82*, *PRB4*, and *MTRNR2L8* were most consistently downregulated.

Biological pathway analysis revealed that the changes in gene expression corresponded to increased keratinization and cell cycle progression/mitosis/cell division. The dysregulated gene expression was identified as most similar to various diseases of the epidermis, including palmoplantar keratoderma, ectodermal dysplasia, ichthyosis, hypotrichosis, and epidermolysis bullosa. The relevant functions of the most highly dysregulated genes, as shown in Table 2, reinforce the similarities to epidermal disease. Most of the upregulated genes have roles in keratinocyte differentiation (*CALML5*, *S100A7*, *KRT16*, *KRT6B*, *KRT14* and *CASP14*) or chronic inflammation (*S100A7*, *KRT16*, *DEFB4A*, *DEFB103A*, and *NPAT*), while most of the downregulated genes have roles in cell cycle regulation and apoptosis (*C11orf65*, *MTRNR2L8*, *PRB3*, *NPAT*, *MTRNR2L12*, *ATM*, and *CASP8*). Retinoic acid (RA)-mediated cell signaling also appeared to be dysregulated, with *ATM* (a serine/threonine kinase that mediates gene expression of RA-dependent genes) being highly downregulated and *CRABP2* (an RA cytosol-to-nuclear shuttle protein) being highly upregulated.

Table 2 also displays drugs that regulate the expression of the most highly dysregulated genes. Since no single drug adequately regulated all the dysregulated genes, we sought to identify FDA-approved drugs that regulated the greatest number of genes. Members of the retinoid family appeared to regulate seven of the upregulated (*CALML5*, *S100A7*, *KRT16*, *DEFB4A*, *KRT14*, *DSG1* and *CRABP2*) and two of the downregulated genes (*ATM* and *CASP8*). Valproic acid regulated three of the upregulated (*CALML5*, *DEFB103A*, and *CRABP2*) and five of the downregulated genes (*C16orf82*, *DPEP1*, *NPAT*, *ATM* and *CASP8*).

## 4. Discussion

Our study reports patterns of gene expression in ITS. RNA sequencing analysis indicated an upregulation of genes and pathways involved in keratinization and chronic inflammation, and downregulation of genes involved in cell cycle regulation. Our results also indicated cellular non-responsiveness to retinoic acid. Subsequently, retinoic acid also emerged as a promising potential therapeutic option for the management of ITS.

To the best of our knowledge, our study is the first to demonstrate the role of dysregulated keratinization in the development of ITS. Keratinization is a process that occurs predominantly in the skin, whereby keratinocytes migrate from the stratum basalis to the stratum corneum to form the keratinized, stratified surface epithelium of the skin. Keratinocytes produce and secrete keratin, a protein that forms intermediate filaments serving structural and barrier-forming roles in the skin. Keratin is also produced by non-skin epithelial cells in the body, such as the cells lining the lumens of the aerodigestive system, where it is involved in cell signaling, cell transport, and maintaining cytoskeletal integrity [13]. However, abnormally increased keratin production (hyperkeratinization) may rarely occur as a result of sustained cellular damage to aerodigestive epithelial surfaces. In the oral cavity, keratinization may occur in reactive lesions (e.g., frictional keratosis, nicotine stomatitis, and hairy leukoplakia), immune-mediated lesions (e.g., lichen planus, and discoid lupus erythematosus), pre-neoplastic and neoplastic diseases (e.g., leukoplakia, proliferative verrucous leukoplakia, and squamous cell carcinoma), and infections (e.g., squamous cell papilloma, verruca vulgaris, and condyloma acuminatum). Esophageal hyperkeratinization may be an uncommon manifestation of chronic gastric acid-induced epithelial damage seen in gastroesophageal reflux disease (GERD) [14,15]. Similarly, tracheobronchial hyperkeratinization has been reported as a possible result of chronic exposure to cigarette smoke or other inhaled environmental pollutants [16]. Squamous metaplasia of the mucociliary airway epithelium, which lends itself to keratinization, has also been described as a potential sequela of high-pressure airway cuffs [17]. Hyperkeratinization processes result in significant alterations to the gross appearance (skin-like lesions) and function of normal epithelial surfaces. In the trachea, this aberrant hyperkeratotic, hyperproliferative tissue can occlude the airway lumen and result in airway stenosis.

The tissue samples in our study also showed evidence of chronic inflammation and dysregulation of cell-cycle and apoptotic pathways. These findings are supported by the previous literature that demonstrates chronic inflammatory processes orchestrated by T-helper 2 cells, M2 macrophages, and fibroblasts [18]. Key molecular players that have been described include TGF-β [19,20], and interleukins-4 (IL-4), -6 (IL-6), and -17 (IL-17) [18]. We found the DEFB103A (defensin β 103A) and DEFB4A (defensin β 4A) genes to be highly upregulated in our samples. β defensins are a family of antimicrobial proteins that provide innate immunity at the aerodigestive mucosal surfaces, and their dysregulated expression may account for persistent chemotactic recruitment of immune cells and chronic inflammation [21]. Cell-cycle dysregulation has also been previously described as a key pathophysiological mechanism driving the hyperproliferative processes observed in ITS [22].

In our search for potential therapeutic for tracheal stenosis, retinoic acid emerged as the top candidate given its ability to regulate several of the most highly dysregulated genes. Retinoic acid is well-known for its role in maintaining epithelial integrity and promoting mucosal immunity [23]. Topical retinoic acid can improve the regeneration of a mucociliary respiratory epithelium after iatrogenic mucosal injury [24]. The genetic alterations in our study point towards a potential loss of adequate cellular response to retinoic acid. ATM (ataxia–telangiectasia mutated) is a serine threonine kinase that is involved in several intracellular processes, including mediating cellular responses to retinoic acid [25]. It follows that the upregulation of CRABP2 (cellular retinoic acid binding protein 2; a cytosol-to-nucleus shuttle protein for retinoic acid) was present as a compensatory mechanism for the cellular non-responsiveness to retinoic acid. Interestingly, retinoic acid itself may help promote the kinase activity of ATM [26]. These findings provide further support for our case for retinoic acid as a therapeutic agent for ITS. Retinoic acid has been used to treat disorders of hyperkeratinization of the skin by virtue of its ability to favorably regulate keratin expression (including KRT6B, KRT14, and KRT16 [27], which were amongst the most highly dysregulated genes in our samples). Moreover, retinoic acid has also been shown to downregulate overexpression of DSG1 (desmoglein-1), which was noted in our samples, and thus promote dissolution of excessive inter-cellular adhesions [28]. Thus, the regulatory acceptance of retinoic acid and its ability to favorably modulate multiple dysregulated pathways substantiate its position as a prime candidate for future testing in ITS disease models.

Our findings of retinoic acid as a promising potential therapeutic agent for the prevention and treatment of tracheal stenosis has exciting implications beyond its use for ITS. Our previous work shows that COVID-19 has the potential to contribute towards the pathophysiology of tracheal stenosis in patients undergoing airway interventions [10]. Interestingly, even with this COVID-19-associated tracheal stenosis, several relevant retinoic acid-related dysregulations are evident. CRABP1 (cellular retinoic acid-binding protein 1: inhibits retinoic acid’s activity) is significantly downregulated, while CRABP2 and RBP1 (retinol-binding protein 1 facilitates enzymatic conversion of retinol to retinoic acid) are significantly upregulated. Enzymatic conversion of retinoid precursors to physiologically active retinoic acid was also suppressed, as evidenced by significant downregulations in RALDH2/ALDH1A2 [10].

Finally, the tracheal stenosis-mitigating effects of retinoic acid can be extended to more experimental applications, such as tracheal graft bioengineering. While these investigational grafts were shown to remain patent and effective beyond the immediate postoperative period, their permanent success was precluded by the development of tracheal granulation tissue causing stenosis and luminal occlusion [29,30,31,32,33,34]. Molecular analysis of the tracheal granulation tissue from the stenotic grafts in the porcine pre-clinical models revealed dysregulations in cellular pathways including inflammation and extracellular matrix remodeling [35]. Further work in this field may reveal a role for retinoic acid to assist in the mitigation and tracheal stenosis in the field of airway bioengineering and transplantation.

This study has limitations. First, this study presented data from only 11 samples. Second, our work discussed the biological plausibility of potential therapeutic options, and future work is required to evaluate actual benefits of these drugs. Third, our work was restricted to genomic analysis; complementary proteomic and histologic analysis would further validate our conclusions.

## 5. Conclusions

Our study demonstrates patterns of gene expression in ITS that indicate an upregulation of genes and pathways involved in keratinization and chronic inflammation, and downregulation of genes involved in cell cycle regulation. Our results also indicated cellular non-responsiveness to retinoic acid due to dysregulations in RA metabolism and signaling. Given the ability of RA to favorably regulate key genes involved in ITS, future studies must explore its efficacy as a potential therapeutic option for patients with recalcitrant ITS.

## Figures and Tables

**Figure 1 biomedicines-12-01323-f001:**
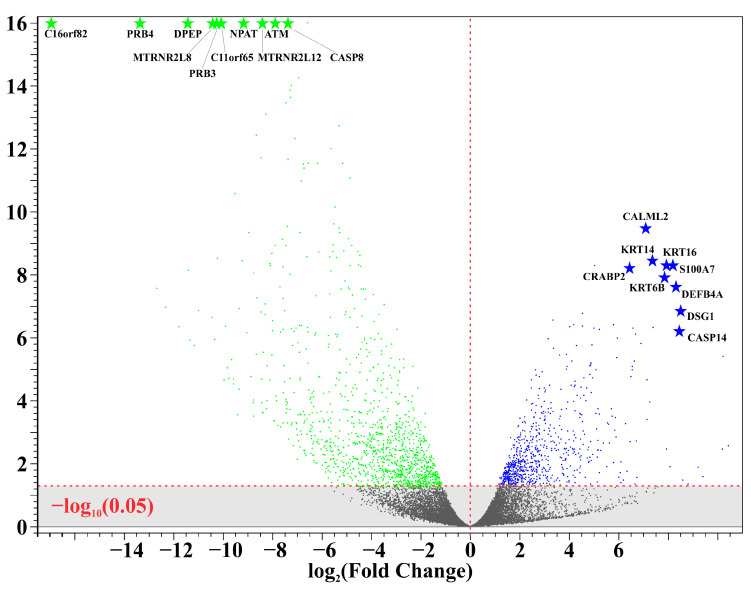
Differential gene expression in iatrogenic tracheal stenosis.

**Figure 2 biomedicines-12-01323-f002:**
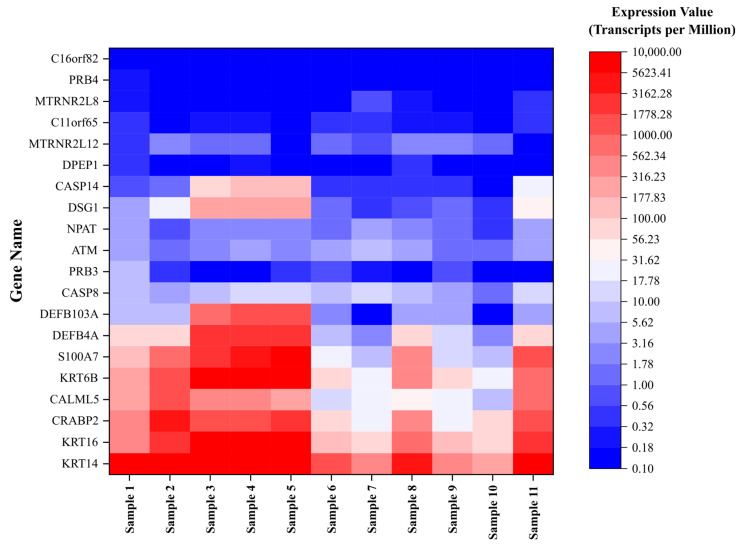
Expression of dysregulated genes in iatrogenic tracheal stenosis.

**Table 1 biomedicines-12-01323-t001:** Dysregulated genes in iatrogenic tracheal stenosis. *ATM*: Ataxia–Telangiectasia Mutated; *C16orf82*: Chromosome 16 open reading frame 82; *C11orf65*: Chromosome 11 open reading frame 65; *CALML5*: Calmodulin-like 5; *CASP14*: Caspase 14; *CASP8*: Caspase 8; *CRABP2*: Cellular Retinoic Acid Binding Protein 2; *DEFB103A*: Defensin beta 103A; *DEFB4A*: Defensin beta 4A; *DPEP1*: Dipeptidase 1; *DSG1*: Desmoglein 1; *KRT16*: Keratin 16; *KRT14*: Keratin 14; *KRT6B*: Keratin 6B; *PRB4*: Proline Rich Protein BstNI Subfamily 4; *MTRNR2L8*: Mitochondrially Encoded 16S RRNA-like 8; *MTRNR2L12*: Mitochondrially Encoded 16S RRNA-like 12; *NPAT*: Nuclear Protein, Coactivator Of Histone Transcription; *PRB3*: Proline Rich Protein BstNI Subfamily 3; *S100A7*: S100 Calcium Binding Protein A7.

Status	Gene Name	Chromosome	Product	Fold Changes	Change Coefficient
Upregulated	*CALML5*	10	Calcium-binding protein	7.09	86.48
	*S100A7*	1	Enzyme inhibitor	7.85	84.86
	*KRT16*	17	Cytokeratin	7.81	84.34
	*DEFB4A*	8	Transporter	8.33	83.46
	*KRT6B*	12	Cytokeratin	7.87	81.33
	*KRT14*	17	Enzyme	7.37	80.98
	*DSG1*	18	Enzyme	8.52	77.89
	*DEFB103A*	8	Enzyme	10.24	76.14
	*CASP14*	19	Enzyme	8.47	71.02
	*CRABP2*	1	Transporter	6.44	68.69
Downregulated	*C16orf82*	16	Plasma protein	−125,487.85	−444.63
	*PRB4*	12	Cytoskeletal protein	−10,386.22	−349.71
	*C11orf65*	11	Cytoskeletal protein and plasma protein	−1174.07	−301.06
	*MTRNR2L8*	11	Plasma protein	−1376.49	−255.23
	*DPEP1*	16	Enzyme	−2730.08	−239.23
	*PRB3*	12	Cysteine rich C-terminal 1	−1210.41	−237.90
	*NPAT*	11	Plasma protein	-572.99	−224.71
	*MTRNR2L12*	3	Cytoskeletal protein and plasma protein	−336.61	−186.51
	*ATM*	11	Enzyme	−235.61	−153.15
	*CASP8*	2	Cytoskeletal protein and plasma protein	−165.63	−152.26

**Table 2 biomedicines-12-01323-t002:** Potential regulatory agents for dysregulated genes. *ATM*: Ataxia–Telangiectasia Mutated; *C16orf82*: Chromosome 16 open reading frame 82; *C11orf65*: Chromosome 11 open reading frame 65; *CALML5*: Calmodulin-like 5; *CASP14*: Caspase 14; *CASP8*: Caspase 8; *CRABP2*: Cellular Retinoic Acid Binding Protein 2; *DEFB103A*: Defensin beta 103A; *DEFB4A*: Defensin beta 4A; *DPEP1*: Dipeptidase 1; *DSG1*: Desmoglein 1; *KRT16*: Keratin 16; *KRT14*: Keratin 14; *KRT6B*: Keratin 6B; *PRB4*: Proline Rich Protein BstNI Subfamily 4; *MTRNR2L8*: Mitochondrially Encoded 16S RRNA-like 8; *MTRNR2L12*: Mitochondrially Encoded 16S RRNA-like 12; *NPAT*: Nuclear Protein, Coactivator Of Histone Transcription; *PRB3*: Proline Rich Protein BstNI Subfamily 3; *S100A7*: S100 Calcium Binding Protein A7.

Status	Gene Name	Relevant Functions	FDA-Approved Agent(s) That Downregulate the Gene	FDA-Approved Agent(s) That Upregulate the Gene
Upregulated	*CALML5*	Terminal differentiation of keratinocytes.	17-β-estradiol Fenretinide Valproic acid	
	*S100A7*	Differentiation of keratinocytes. Inhibits chronic inflammation and remodeling		17-β-estradiol Isotretinoin/Alitretinoin/Tretinoin/4-oxoretinoic acid Calcitriol
	*KRT16*	Differentiation of keratinocytes. Promotes chronic inflammatory response (chemotactic for monocytes and lymphocytes)	Fenretinide/Isotretinoin Calcipotriol Folic acid Ivermectin Acetaminophen	Calcipotriol Cisplatin 17-β-estradiol 5-fluorouracil Gentamycin Metronidazole
	*DEFB4A*	Promotes inflammation as part of antimicrobial response	Tretinoin	17-β-estradiol Progesterone
	*KRT6B*	Differentiation of keratinocytes. Antiviral	17-β-estradiol	Calcitriol Progesterone 17-β-estradiol
	*KRT14*	Differentiation of keratinocytes. Breakdown of extracellular matrix in tissue remodeling	Folic acid Isotretinoin Ivermectin Acetaminophen Dexamethasone Methimazole	17-β-estradiol Tretinoin
	*DSG1*	Cell–cell adhesion Antimicrobial and antiviral	17-β-estradiol Retinoic acid	-
	*DEFB103A*	Promotes inflammation as part of antimicrobial response	Valproic acid	Folic acid
	*CASP14*	Differentiation of keratinocytes.	Folic acid Acetaminophen	-
	*CRABP2*	Retinoic acid cytosol-to-nuclear shuttling (facilitates RA activity)	17-β-estradiol Dexamethasone Valproic acid	Retinoic acid Gentamycin Valproic acid
Downregulated	**Gene Name**	**Relevant Functions**	**FDA-approved Agent(s) that Downregulate the Gene**	**FDA-approved Agent(s) that Upregulate the Gene**
	*C16orf82*		Valproic acid	
	*PRB4*			
	*C11orf65*	Negative regulator of cell proliferation		Levofloxacin
	*MTRNR2L8*	Negative regulator of cell proliferation		
	*DPEP1*	Pro-inflammatory	Cilastatin Gentamicin Methimazole Mifepristone Acetaminophen Valproic acid	Dexamethasone Doxorubicin Progesterone Vancomycin
	*PRB3*	Antimicrobial Pro-apoptotic	17-β-estradiol	17-β-estradiol
	*NPAT*	Cell-cycle regulation Anti-inflammatory and anti-microbial for mucosal surfaces	Acetaminophen Valproic acid	Valproic acid
	*MTRNR2L12*	Anti-apoptotic		
	*ATM*	Mediates cellular response to retinoic Acid DNA damage response (cell cycle regulator)	Tretinoin 17-β-estradiol Calcitriol Cisplatin Doxorubicin Rifampicin Valproic acid	17-β-estradiol Doxorubicin Gentamycin Acetaminophen Valproic acid
	*CASP8*	Pro-apoptotic	17-β-estradiol 5-fluorouracil Doxorubicin	17-β-estradiol 5-fluorouracil Alitretinoin/Isotretinoin Cisplatin Dexamethasone Diclofenac Doxorubicin Gentamicin Methotrexate Mitomycin Paclitaxel Acetaminophen Progesterone Valproic acid Vincristine

## Data Availability

The data presented in this study are available on request from the corresponding author. (Dataset contains confidential genetic data from patients).

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
