# Peer review of "Identifying Molecular Pathophysiology and Potential Therapeutic Options in Iatrogenic Tracheal Stenosis"

_biomedicines, 2024, doi:10.3390/biomedicines12061323_

Round 1

Reviewer 1 Report

Comments and Suggestions for Authors

The authors describe a basic science investigation of gene expression of tracheal granulation tissue in patients with iatrogenic airway injury.  Prior basic science work in this space has primary focused on protein expression on cellular level via immunohistochemical analysis and flow cytometry as well as analysis of the local immune environment.  This is the first study to this reviewer's knowledge looking comprehensively at RNA expression of tracheal granulation tissue.  The authors utilize the results to look for potential therapeutic agents, which provides additional value to the manuscript.  The manuscript is well written without notable grammatical errors.

The introduction is concise and references well the extant literature on post-intubation airway injury, iatrogenic airway stenosis, treatments, and existing literature on proposed pathogenic mechanisms.

Specimens were collected from patients with iatrogenic airway stenosis.   I would recommend the authors specify more about enrollment criteria.  They state "severe, recalcitrant ITS".  Is severity based off degree of luminal compromise, degree of symptoms, or both?  Recalcitrant implies need for re-operation.  Were all of these patients enrolled after they had recurrence despite first endoscopic treatment?

All specimens were stated to be of tracheal granulation and not mature fibrosis.  This is an appropriate methodology decision, though the authors may want to comment on justification for this choice (easier tissue to obtain?  Granulation implies looking are the earlier mechanisms of injury and initial inflammatory response?).  The number of specimens is low, but consistent with most research of this nature looking at iatrogenic airway stenosis, owing to the rarity of the clinical condition and costs of tissue analysis.

The authors used a commercial platform/service for RNA analysis. The specifics of methodology are sparse, though it would be worth comment of this is due to the process being proprietary.

The methodology of gene expression analysis is appropriate.  The figures and tables are excellent.

The discussion reaches appropriate conclusions.  The authors appropriate integrate extant literature on the immune microenvironment and wound healing into their conclusions.  The identification of a potential therapeutic target with a favorable toxicity profile is exciting and as they state in limitations, warrants additional investigation.

Author Response

SUMMARY OF KEY MANUSCRIPT CHANGES TO ADDRESS REVIEWER 1 COMMENTS

Comment

Revised Text

1

Section “2.1. Patient Enrollment”: lines 89-91

2

Section “2.1. Patient Enrollment”: lines 93-94

3

Section “2.2. RNA Sequencing of Samples”: lines 101-106

DETAILED RESPONSE TO REVIEWER 1 COMMENTS

We greatly appreciate the reviewer’s attention to detail, kind words, and the opportunity to address their comments. To follow up on their feedback, we have revised the body of the manuscript to clarify ambiguous points and provide congruence with journal/publisher guidelines for article publication.

Key revisions have been highlighted in the manuscript for ease of reference using the following scheme

  • Text additions have been marked in red
  • Removed text has been marked with strikethrough

In addition, the manuscript has been adapted to the journal proofing format.

COMMENT 1: I would recommend the authors specify more about enrollment criteria.  They state "severe, recalcitrant ITS".  Is severity based off degree of luminal compromise, degree of symptoms, or both?  Recalcitrant implies need for re-operation.  Were all of these patients enrolled after they had recurrence despite first endoscopic treatment?

RESPONSE 1: To provide further clarification into patient selection criteria as suggested by the reviewer, we have included a sentence to specify the degree of luminal compromise (>70%) in “severe” ITS and requisite prior procedures (at least two) to qualify as “recalcitrant”.

Key text revisions include:

  • Section “1. Patient Enrollment and Sample Collection
    • Inclusion criteria include over 70% tracheal lumen occlusion and at least two recurrences which required cryotherapy/balloon dilation procedures.

COMMENT 2: All specimens were stated to be of tracheal granulation and not mature fibrosis.  This is an appropriate methodology decision, though the authors may want to comment on justification for this choice (easier tissue to obtain?  Granulation implies looking are the earlier mechanisms of injury and initial inflammatory response?). 

RESPONSE 2: Since we are looking to prevent and treat active tracheal stenosis, we focused our efforts on earlier disease mechanisms prior to scarring (where disease progression and tissue remodeling may be irreversible), as postulated by the reviewer. Granulation tissue represents these active disease processes associated with tracheal stenosis, hence our decision to include this specific tissue in the study. To provide clarity to this underlying reasoning, we have made modified the manuscript to make this point more explicit.

Key text revisions include:

  • Section “1. Patient Enrollment and Sample Collection
    • To understand active disease mechanisms in ITS (prior to potentially irreversible tissue scarring), we examined transcriptomic dysregulation in granulation tissue.

COMMENT 3: The specifics of methodology are sparse, though it would be worth comment of this is due to the process being proprietary.)

RESPONSE 3: Additional information about the ordered service from the vendor (Azenta Life Sciences) has been included to improve methodological clarity as per reviewer suggestion.

Key text revisions include:

  • Section “2. RNA Sequencing of Samples
    • Frozen tissue samples were sent to Azenta Life Sciences for RNA extraction, processing, and RNA sequencing. cDNA libraries were generated using Illumina preparation kits with Poly(A) selection. Sequencing was conducted using the Illumina HiSeq® platform configured to the following settings: 2x150bp, ~350M paired-end reads, single index, per lane with ≥80% of bases ≥Q30. Analysis was performed using the vendor standard RNA-seq analysis.

We hope that these revisions have elevated the quality of the manuscript to the publication standards of the reviewer and Biomedicines

Reviewer 2 Report

Comments and Suggestions for Authors

Thank you very much for inviting me to review this work. This is a very interesting manuscript about exploring genetic profiles of patients with recalcitrant ITS to understand underlying pathophysiology and identify novel therapeutic options. 

The introduction is written in a comprehensive manner, the methods are also described in detail and I have no objections to them. The results are presented very well and correspond to the conclusions given by the authors. The discussion is also very well described and exhaustive of the topic (as far as current knowledge is concerned). It is a very good manuscript, and my only comment is the need to edit the form of references in the text (superscripts instead of providing them in the usual form in the text) - substantively, the manuscript is very good and I congratulate the authors on both the wonderful research and the manuscript - the only thing that is necessary is editing text in accordance with the requirements of the magazine and publishing house.

Author Response

SUMMARY OF KEY MANUSCRIPT CHANGES TO ADDRESS REVIEWER 2 COMMENTS

Comment

Revised Text

1

Section “1. Introduction”: lines 47, 48, 49, 51, 57, 58, 61, 69

Section “2.4. Data Analysis”: lines 123, 125

Section “4. Discussion”: lines 203, 211, 213, 215, 223, 224, 229, 231, 235, 236, 239, 243, 246, 249

DETAILED RESPONSE TO REVIEWER 2 COMMENTS

We greatly appreciate the reviewer’s attention to detail, kind words, and the opportunity to address their comments. To follow up on their feedback, we have revised the body of the manuscript to clarify ambiguous points and provide congruence with journal/publisher guidelines for article publication.

Key revisions have been highlighted in the manuscript for ease of reference using the following scheme

  • Text additions have been marked in red
  • Removed text has been marked with strikethrough

In addition, the manuscript has been adapted to the journal proofing format.

COMMENT 1: It is a very good manuscript, and my only comment is the need to edit the form of references in the text (superscripts instead of providing them in the usual form in the text)… the only thing that is necessary is editing text in accordance with the requirements of the magazine and publishing house.

RESPONSE 1: The in-text reference citations have been modified to be in concordance with publisher MDPI and journal Biomedicines requirements. In addition, the manuscript has been adapted to the Biomedicines journal proof format.

Since there are a substantial number of similar revisions associated with this reviewer comment, an example revised in-text reference citation is included here.

  • Tracheal stenosis occurs most commonly as a sequela of iatrogenic causes such as prolonged intubation and tracheostomy (1).

We hope that these revisions have elevated the quality of the manuscript to the publication standards of the reviewer and Biomedicines.
